# Molecular Characteristics of Amyloid Precursor Protein (APP) and Its Effects in Cancer

**DOI:** 10.3390/ijms22094999

**Published:** 2021-05-08

**Authors:** Han Na Lee, Mi Suk Jeong, Se Bok Jang

**Affiliations:** 1Department of Molecular Biology, College of Natural Sciences, Pusan National University, Jangjeon-dong, Geumjeong-gu, Busan 46241, Korea; emille96@naver.com; 2Insitute for Plastic Information and Energy Materials and Sustainable Utilization of Photovoltaic Energy Research Center, Pusan National University, Jangjeon-dong, Geumjeong-gu, Busan 46241, Korea

**Keywords:** amyloid precursor protein, Alzheimer’s disease, cancer, proliferation, migration

## Abstract

Amyloid precursor protein (APP) is a type 1 transmembrane glycoprotein, and its homologs amyloid precursor-like protein 1 (APLP1) and amyloid precursor-like protein 2 (APLP2) are highly conserved in mammals. APP and APLP are known to be intimately involved in the pathogenesis and progression of Alzheimer’s disease and to play important roles in neuronal homeostasis and development and neural transmission. APP and APLP are also expressed in non-neuronal tissues and are overexpressed in cancer cells. Furthermore, research indicates they are involved in several cancers. In this review, we examine the biological characteristics of APP-related family members and their roles in cancer.

## 1. Introduction

Amyloid precursor protein (APP) is a type 1 transmembrane glycoprotein that plays important roles in neural transmission and neuronal homeostasis and development [1]. APP is a member of the APP-related protein family that includes amyloid precursor-like proteins 1 and 2 (APLP1 and APLP2) in mammals and amyloid precursor protein-like (APPL) in *Drosophila* [2]. APP has 695 to 770 isoforms and the homologies of APP and APLP are highly conserved. Most of the research on APP has been conducted in the context of Alzheimer’s disease (AD) and the processing of APP to amyloid β. However, APP is highly expressed in brain and other organs and is overexpressed with APLP in multiple cancers, including glioblastoma and breast, pancreatic, lung, colon, and prostate cancer [3]. Furthermore, APP and APLP are known to participate in the progression, proliferation, and migration of cancers cells.

## 2. Amyloid Precursor Protein

Over the past 20 years, it has become clear that amyloid precursor protein (APP) is an important player in Alzheimer’s disease (AD), a common age-related neurodegenerative disorder that causes progressive memory and cognitive impairments. Pathologically, the characteristics of AD are neurofibrillary tangles and senile plaques comprised of amyloid-β deposits in the cerebral cortex [4]. APP is a member of the APP-related protein family, which includes APL-1 in Caenorhabditis elegans, amyloid precursor-like proteins (APLP1 and APLP2) in mammals, and the amyloid precursor protein-like (APPL) in Drosophila [2]. The amyloid-like molecules lack an Aβ domain but possess a conserved motif, which includes an extracellular and an intracellular domain, that exhibits high sequence identity [5]. APP is synthesized in the central nervous system and is then transported to other locations through axons of the peripheral nervous system [6]. APP is highly enriched in the brain and may like APLP2 be detected in other tissues. The human APP gene is located on chromosome 21 and exists in multiple isoforms due to alternative splicing [7]. Moreover, the human APP gene has 18 exons, and alternative splicing of APP results in eight isoforms comprised of 695 to 770 amino acids [2]. Furthermore, APP mutations are associated with amyloid-β (Aβ) deposition and the pathogenesis of Alzheimer’s disease (AD). Approximately 25 mutations in the APP gene have been reported to be pathogenic and to result in amino acid substitutions within flanking the Aβ domain [8].

### 2.1. Amyloid Precursor Protein Structure

All APP structures consist of an extracellular domain, an Aβ domain, and a cytoplasmic region. To understand APP function, one requires detailed knowledge of its basic structure and of the functions of specific regions. In APP 770, the N-terminal (residues 18–123) contains a growth factor-like domain (GFLD) that consists of nine β-strands and one α-helix. This domain is cysteine-rich and has two disulfide knots that are well-conserved in APP and APLP. A disulfide bridge is formed between Cys98 and Cys105 and is stabilized by a β-hairpin loop consisting of basic residues that contribute to its positive charged surface [9]. In a previous study, it was suggested that the N-terminal domain of APP might stimulate neurite outgrowth [10], which suggests that the N-terminal domain may be responsible for the growth factor-like function of APP. The copper-binding domain (CuBD) of APP occupies residues 124–189 of APP and consists of an α-helix packed with a triple-stranded β-sheet. Notably, His 147, His 151, Tyr 168, and Met 170 are tetrahedrally rearranged and compose the copper-binding site [9]. GFLD and CuBD are referred to as the extracellular 1 region, which is linked to the acidic domain at 190–289. In APP and APLP, another extracellular domain called the bovine pancreatic trypsin inhibitor (BPTI)/Kunitz protease inhibitor domain follows at residues 290–364, and this domain is rich in aspartate and glutamate residues [3]. The extracellular 2 region positioned at 365–575 consists of a RERMS sequence and a central APP domain (CAPPD). The extracellular and transmembrane (TM) regions are positioned at 576–699. The amyloid β region is positioned at 672–712/713 [11]. The intracellular domain contains a highly conserved sequence that is crucial for the functional regulation of APP; it is believed to mediate interactions with other proteins that lead to the productions of transcriptional factors and initiate nuclear signaling (Figure 1).

### 2.2. Amyloid Precursor Protein Modifications

Methionine at position 35 (Met35) is the oxidation site of amyloid β, and in AD patients or the cognitively impaired the brain is exposed to oxidative stress. According to reports, oxidation of Met35 interrupts the formation of amyloid β protofibrils and fibrils and inhibits aggregation in vitro [12]. The phosphorylation sites of amyloid β are located at serine in positions 8 and 26 and at tyrosine in position 10. Phosphorylation of serine residue 26 is mediated by cdc2 kinase, and it was recently reported cdc2 phosphorylates Aβ in human NT2 neurons and AD brain samples, and that a cdc2 kinase inhibitor reduced the neurotoxic effect of Aβ in NT2 neurons [13]. Phosphorylation of serine residue 8 was localized to amyloid plaque under pathological conditions, and phosphorylation of ser8 promoted the formation of oligomeric Aβ aggregates [14]. Nitric oxide (NO) mediates the formation of S-nitrothiols at cysteine residues and the nitration of dityrosine at tyrosine residues. NO is a free radical and functions as a diffusible neurotransmitter. High concentrations of NOS2-derived NO result in the formation of reactive peroxynitrite due to the nitration or nitrosylation of amino acid residues [15]. In AD, NOS is upregulated and nitrates tyrosine 10 of Aβ, which enhances Aβ aggregation. Furthermore, Aβ nitration was found to induce the formation of amyloid plaque in APP/PS1 mice and resulted in early phase AD [16].

### 2.3. Amyloid Precursor Protein Processing

APP processing can occur via a non-amyloidogenic or an amyloidogenic pathway (Figure 2) and occurs at cell surfaces and in trigeminal neuralgia, in which APP is transported to cell surfaces and endosomal compartments [2]. APP is first cleaved within its extracellular domain by BACE β-secretase (a membrane-bound aspartyl protease) within the N$H_{2}$ terminus of Aβ to release sAPPβ, a 100 kD soluble N$H_{2}$ terminus fragment, and C99, a 12 kD COOH terminal fragment or C100 [17]. BACE is localized to trans-Golgi networks (TGNs) and endosomes, in which APP cleavage occurs. Moreover, APP cleavage by BACE takes place within Aβ at asp1 (a β-site) and between tyr10 and glu11 ($\beta^{\prime }$-site). APP fragmented by BACE is subsequently cleaved within the APP transmembrane domain by γ-secretase, the activity of which is regulated by intramembrane proteolysis (RIP) and requires presenilin (PS1 and PS2) and nicastrin to form γ-secretase complex [4]. PS is expressed in most cell types during development (PS1 and PS2), which share 65% identity [18]. PS possess hydrophobic regions (HRs) that consist of six to eight transmembrane domains (TMs). PS performs endoproteolysis using a cytoplasmic loop that connects to other proteins, which can include APP, Notch, or β-catenin [18]. Moreover, presenilin (PS1 and PS2) and nicastrin are cleaved by APP and Notch receptors, which arise from the Notch intracellular signaling domain (NICD). Furthermore, E-cadherin, ErbB-4 receptor tyrosine kinase, and LDL receptor-related protein (LPR) have been reported to be cleaved by presenilin [4]. Thus, cleavage by the γ-secretase activity of C99 APP fragment results in the release of both Aβ40 and Aβ42 and APP intracellular domain (AID or AICD). Aβ40 constitutes 90% of Aβ secreted in brain, and Aβ42 accounts for ≈10% [8]. Processing by α-secretase can release the soluble N-terminal derivative sAPPαand C83. α-Secretase cleavage is followed by γ-secretase cleavage, which results in APP non-amyloidogenic extracellular fragments (p3) and AICD. α-Secretase cleavage occurs within Aβ peptide at position 16–17. Moreover, approximately 30% of APP occurred α-secretase cleavage processing that is highly more than β-secretase cleavage processing [4]. There are two different types of α-secretases, namely, zinc metalloproteinases, such as TACE/ADAM17, ADAM9, ADAM10, and MDC-9, and aspartyl protease BACE2 [8]. The α-secretase site of APP is positioned between ${\mathrm{Lys}}^{16}$ and ${\mathrm{Leu}}^{17}$ at the Aβ domain [8]. In cancer, APP and APLP cleavage involve hormone insulin-like growth factor 1 (IGF-1), which is a well-known regulator of cell growth and cancer progression. IGF-1 and IGF-1 receptor are expressed in brain and transported through the blood–brain barrier. Recent studies have implicated IGF-1/insulin in AD as a promoter of Aβ production via the secretase pathway involving APP phosphorylation [19]. Conversely, in human neuroblastoma, IGF-1 increases the secretions of sAPPα, sAPLP1, and sAPLP2 and reduces Aβ production [20].

### 2.4. Amyloid Precursor Protein Trafficking

The trafficking of APP may occur secretion of Aβ peptides which is uniformly expressed in the neuron, but Aβ deposition has a severe effect within extracellular and intracellular. APP synthesized in the endoplasmic reticulum (ER) is transported through the secretory pathway, which arises from ER and passes through the Golgi/trans-Golgi network (TGN) to the plasma membrane. Trafficking of APP through the secretory pathway requires modification by N- and O-glycosylation, ubiquitination, phosphorylation, and tyrosine sulfation [21]. In particular, APP glycosylation occurs in different regions, N-glycosylation occurs in the ER, and O-glycosylation in the TGN [22]. APP has two glycosylation sites, Asn467 and Asn496. Deletion of Asn467 and Asn496 in CHO cells leads to reduced APP secretion, which suggests APP glycosylation is affected by the intracellular sorting of APP [23]. For APP to be transported to the plasma membrane, APP is packaged in post-Golgi transport vesicles with the involvement of heterotetrameric adaptor protein complex AP4, which interacts with a cytosolic sequence of APP [21]. In the plasma membrane, APP is either cleaved by α-secretase or internalized within clathrin-coated vesicles [24]. APP localized in endosomes is either recycled to cell surfaces or delivered to lysosomes for degradation. In the Golgi network, both α-secretase and β-secretase can encounter APP, leading to occur a part of APP proteolysis [21]. Approximately ≈10% of nascent APP is present in the plasma membrane and most of them localizes in TGNs in the steady state. In the plasma membrane, APP that is not proteolytically processed undergo endocytosis within minutes of reaching the cell surface, but rather is internalized by TGN using its YENPTY internalization motif near its C-terminus (residues 682–687 of the APP695 isoform) [24]. The conserved YENPTY motif is a bound cytosolic factor (intracellular adaptor), such as APP-binding A (APBA, previously termed MINT), APP-binding B1 (APBB1, previously termed Fe65), RAB GTPase, and sorting nexins [21]. Moreover, the trafficking of APP can be regulated by low-density lipoprotein receptors about apolipoprotein E (ApoE), which are associated with the onset of AD. Phosphorylations of APP at S655 and T668 (based on APP695 numbering) are affected by APP trafficking. According to research, mature APP is modified by phosphorylation in its cytoplasmic region [22].

## 3. Amyloid Precursor Protein in Cancer

APP is involved in cell survival, cellular adhesion, differentiation, and migration [25], and the expressions of APP and APLP are abundant in various non-neuronal tissues, which suggests they are involved in tissue growth. According to some studies, the expressions of APP and APLP differ in many types of cancers. It has been reported to be correlated with several human malignant cancers, e.g., colon, pancreatic, lung, parathyroid, breast, thyroid, and prostate cancer [26]. According to APP expression studies, APP silencing reduced cancer proliferation, and it was suggested that APP knockdown in breast cancer reduces cell growth via cyclin-dependent kinase inhibitor (p. 27) induction, which indicated it promotes apoptosis and increases treatment sensitivity [27].

### 3.1. Amyloid Precursor Protein in Breast Cancer

APP has been suggested to be oncogenic in breast cancer, which is a significant cause of morbidity and mortality among women [28]. Furthermore, breast cancer has a high metastatic rate, which results in high invasiveness and recurrence [29]. Breast metastases tend to occur in an organ-specific manner, which has been attributed to high expressions of cell adhesion molecule. According to research, epidermal growth factor receptor overexpression in breast cancer can influence the central nervous system [30]. Silencing-mediated knockdown of APP in tumor cells resulted in normal proliferation and cell survival and basal migratory activity [31]. The intracellular domain of APP influences the physiological functions of nerves and immune cell migration and invasion by interacting with adaptor proteins and activating downstream intracellular signaling molecules [32]. Moreover, the overexpression of APP in MDA-MB-231 cells increased levels of mesenchymal markers, including MMP-9, MMP-2, MMP3, N-cadherin, and vimentin, but decreased those of epidermal-associated markers, including cytokeratin, which implies APP overexpression in association with the mitogen-activated protein kinase (MAPK) signaling pathway is implicated in breast cancer invasiveness and metastasis [28]. According to a previous study, the intracellular domain of APP could activate the MAPK signaling pathway [33]. Moreover, the overexpression of APP through the MAPK signaling pathway upregulates the phosphorylation levels of MAPK signaling pathway components, including MLK3, MEK4, and JNK3 [28].

Breast cancer is divided different subtypes in expression of estrogen receptor (ER), progesterone receptor (PgR), and human epidermal growth factor 2 (HER2). Androgen receptor (AR) is expressed in more than 60% of breast cancer and 90% of ER-positive tumors [34]. APP immunoreactivity has been detected in 49 percent of breast carcinoma tissues, and increases in this percentage are associated with androgen receptor in ER-positive breast carcinoma. Moreover, APP mRNA was reported to be induced by bioactive androgen dihydrotestosterone (DHT) in MCF-7 cells [26]. Although APP is positively associated with androgen receptor, a recent study reported a marginal association between APP and AR expression in luminal A cancer. In this study, APP expression was highly expressed in non-luminal breast cancer, which demonstrated human epidermal growth factor receptor 2–overexpressed (HER2-OE, 39.3%) and triple-negative breast cancer (TNBC, 38.2%), as compared with 8% in luminal A cancer. Additionally, APP was found to be an independent poor prognostic factor in non-luminal cases in TNBC [35]. Likewise, knockdown of APP expression reduced breast cancer cell migration and cell growth. Moreover, s${\mathrm{APP}}_{\alpha }$ presented similarly, and inhibition of ADAM10 blocked s${\mathrm{APP}}_{\alpha }$ expression and suppressed cell growth and migration [25].

### 3.2. Amyloid Precursor Protein in Glioblastoma

Glioblastoma (GBM) is the most aggressive and invasive primary tumor of the central nervous system and the most malignant glioma found in the elderly. Median survival of GBM is only 15 months, and the disease is considered incurable [20]. Despite the considerable efforts made to devise an effective therapy, GBM continues to be associated with a dismal prognosis [36]. Recently, studies on genetic alterations in GBM determined three major pathways, that is, a receptor tyrosine kinase (RTK) pathway, a p53 pathway, and a retinoblastoma protein (RB) tumor suppressor pathway, which are involved in the pathogenesis of GBM [37]. Furthermore, cyclooxygenase-2 (COX-2), cytosolic phospholipase, interleukin-1β (IL-1β), and the β-amyloid precursor protein of proinflammatory and neurodegenerative gene were found to be upregulated in the American Tissue Culture Collection of glioma and glioblastoma. In AD, these genes link inflammatory signaling cascades, and gliosis. Therefore, increasing βAPP and proinflammatory expression in glioma and glioblastoma would understand βAPP–COX-2–CPL–IL-1β signaling and provide a therapeutic strategy [38]. In addition, GBM is positively associated with mortality in Alzheimer’s disease [39]. Immunostaining revealed that amyloid-β (1–42) deposition is observed in glioma tumors and nearby blood vessels in glioma mouse models. Thioflavin can be used to identify the aggregated form of amyloid-β in glioma tumors [40], and amyloid precursor-like protein 2 (APLP2) is also closely related to glioblastoma. Previous reports have shown APLP2 is involved in cell invasion, cell–cell adhesion, proliferation, signaling, and wound healing [41,42,43], and that disorders of APLP2 are involved in metastasis, cell growth, and invasion in multiple cancers, including breast cancer, pancreatic cancer, lung cancer, and colon cancer [26,44].

### 3.3. Amyloid Precursor Protein in Nasopharyngeal Carcinoma (NPC)

Nasopharyngeal carcinoma (NPC) is common cancer in Southern China and Southeast Asia and is associated with herpesvirus infection in epithelial malignancies caused by continuous exposure to smoke, heavy metals, or dust. Chemotherapy provides an effective therapeutic strategy, but 20% of patients that receive radiotherapy experience recurrence. NPC is usually diagnosed on the basis of imaging or biopsy findings [45]. In CNE-2 cells (an NPC cell line), APP was upregulated by transforming growth factor-α (TGF-α) stimulation but had pretreatment with epidermal growth factor receptor (EGFR) tyrosine kinase inhibitor [46]. Moreover, APP expression is upregulated in NPC tissues, and it was reported that APP levels were increased in NPC patients treated by radiotherapy and suggested that APP might be a useful diagnostic and prognostic biomarker of response to radiotherapy in NPC [45]. APP is also involved in the invasion and migration of nasopharyngeal carcinoma cells. In one study, APP knockdown inhibited the invasion and migration of human nasopharyngeal carcinoma cell line (SUNE-1) and increased the expressions of metastasis suppressor-related genes. However, silencing of APP reduced the mRNA expressions of metastasis and progression-related genes. Moreover, APP silencing was suppressed factor that involved epithelial–mesenchymal transition (EMT). Furthermore, APP knockdown decreased the protein expressions of mitogen-activated protein kinase (MAPK) signaling pathway-related genes, suggesting its involvements in invasion and migration in NPC [47].

### 3.4. Amyloid Precursor Protein in Pancreatic Cancer

Characteristically, pancreatic ductal adenocarcinoma is associated with extensive invasion and a propensity to metastasize, and thus its prognosis and survival rate are poor [48]. According to previous research, amyloid precursor-like protein 2 (APLP2) is highly expressed in pancreatic cancer [49]. Moreover, APLP2 shares homology with amyloid precursor protein and is involved in cell migration, signaling, and proliferation [43,50], and binds with major histocompatibility complex (MHC) class I molecules to trigger viral or tumor antigen production by T lymphocytes [51]. Beta 2-microglobulin (β-2M)-associated MHC class I heavy chains present antigenic tumor or pathogen peptides in cytotoxic T cells. The expression of β-2M is maintained or elevated in renal cell carcinoma, oral squamous cell carcinoma, breast cancer, and prostate cancer, and studies have shown β-2M/HLA class I heavy chain/peptide complex binds APLP2 in PANC-1 and S2-013 cells that are involved in migration. Furthermore, it was found that pancreatic cancer migration diminished after β-2M knockdown, which also downregulated the expressions of APLP2 in PANC-1 and S2-013 [52]. A recent study reported APLP2 C-terminal fragment and APLP2-modified glycosaminoglycan are highly expressed in pancreatic cancer cell lines (S2-013, BxPC3) as compared with APP full-length and C-terminal fragments [49]. Moreover, the expression of APLP2 was found to be upregulated by human papillomavirus E6 and E7 oncogenes, as well as K-RasG12D, and SV40 small t antigen expression were upregulated in the pancreatic ductal cells. Knockdown of APLP2 and APP were observed to reduce pancreatic cancer cell growth by more than 60%, and S2-013 cells treated with a β-secretase inhibitor prevented the formation of APLP2 C-terminal fragments and decreased growth [49]. Actin structure is also influenced by cancer cell division and proliferation. Downregulation of APLP2 in pancreatic cancer changed actin cytoskeletons, diminished migration and invasion, and restricted metastatic spread. Furthermore, APLP2 affected actin structure and contributed to metastasis [44].

### 3.5. Amyloid Precursor Protein in Hepatocellular Carcinoma

Liver cancer also has high malignancy, mortality, and morbidity. Hepatocellular carcinoma (HCC) is of serious global concern [53], and its factors are viral hepatitis and immoderate alcohol intake [54]. Preferred therapeutic strategies involve surgery and chemotherapy but drug resistance development is common. Drug resistance is associated with the overexpression of drug resistance proteins [55] and regulations of cellular signaling pathways [56]. 5-Fluorouracil (5-FU) is a widely adopted anti-cancer drug that targets nucleoside metabolism and causes cell death. However, despite their many advantages, 5-FU-based therapies are prone to drug resistance [57]. According to a study on a 5-FU-resistant HCC cell line, APP levels were significantly overexpressed in resistant HCC cells. Interestingly, 5-FU also inhibited the proliferation of liver cells but not APP-overexpressing liver cells. In addition, the expression levels of apoptosis-related genes were suppressed in APP overexpressing liver cells, and this downregulation inhibited activation of the apoptosis pathway, which was related to regulation of the mitochondrial apoptosis pathway. APP may regulate the expression level of mitochondrial pathway-related protein through activation of the MAPK signaling pathway [53]. In hepatocellular carcinoma, the regulation of amyloid precursor protein is associated with histone deacetylase. Aberrant epigenetic modification is related to the development of cancer, and histone acetyltransferase and histone deacetylase (HDAC) importantly contribute to gene regulation. Moreover, histone deacetylases are associated with the development of many cancers. Knockdown of HDAC1 resulted in APP downregulation, caspase-7 cleavage, and apoptosis, which demonstrates an association between APP expression and the regulation of HDAC1 [58].

### 3.6. Amyloid Precursor Protein in Colon Cancer

Colorectal cancer (CRS) is the third most common malignancy and an important cause of cancer-related mortality despite the existence of a number of therapeutic options [59]. Many strategies have been used to treat colon cancer, but in most cases the disease progresses. Carbamazepine (CBZ) is an antiepileptic drug and is also used to treat children with liver disease. CBZ inhibits sodium and potassium channel activities [60], and recently was reported to have effects similar to histone deacetylase inhibitors [61]. APP concentration with the treatment of CBZ in human colon cancer diminished. Nalproic acid (VPA), a similar antiepileptic drug, has also been reported to have anti-cancer effects. In human colon cancer, VPA treatment was found to reduce APP concentration [62].

### 3.7. Amyloid Precursor Protein in Prostate Cancer

Prostate cancer (PC) is the second most diagnosed malignancy in men—a total of 1.2 million PC cases and 0.3 million deaths were reported worldwide in 2018 [63]. The transformation of prostate cancer is associated with the transition of normal prostate tissue to prostatic intraepithelial neoplasia. Prostate cancer is treated by surgical resection and radiotherapy, but subsequent disease recurrence usually leads to metastasis [64]. Androgen deficiency therapy is often used to treat metastatic prostate cancer. Prostate cancer expresses androgen receptors, and androgen signaling is essential for prostate tumor growth [65]. APP has been reported to be associated with androgen-responsive genes in prostate and breast cancer [66], and it has been suggested APP promotes malignancy [67]. Studies have shown APP regulates the proliferation and migration of prostate cancer cells. Microarray analysis showed that the expression level of ADAM metalloproteinases and EMT-related genes reduced that LNCaP and DU145 cells were transfected with siAPP but overexpression of APP increased migratory activity and upregulated the expressions of EMT-related genes and ADAM metalloproteinases [67].

## 4. Conclusions

APP plays important roles in physiological functions, such as neuronal homeostasis, development, and neural transmission. In AD, amyloid plaque causes pathological situations associated with defects in neurons and neural transmission. However, despite enormous research efforts, the precise cause of AD has not been identified. Amyloid plaque is generated by the amyloidogenic pathway, and in this pathway, APP is cleaved by beta-secretase and subsequently by γ-secretase. Although APP is expressed in neuronal tissues, APP and APLP are both expressed in various non-neuronal tissues. Furthermore, APP and APLP have been reported to influence several cancers, although the signaling pathways responsible are not understood. In some cancers, APP and APLP regulate proliferation, migration, and disease progression (Figure 3). Studies have shown that APP signaling is mediated by the transcription coactivator YAP [68], which will undoubtedly provide clues about the nature of APP and APLP-related signaling. Besides APP and APLP, another amyloidogenic protein such as α-synuclein (SNCA) and transthyretin (TTR) is reported to an association with tumorigenesis and tumor growth. TTR is a tetrameric protein composed of 127 amino acid residues and is synthesized in the liver and choroid plexus of the brain. This protein is occurred single amino acid substitutions, leading to familial amyloidotic polyneuropathy (FAP) [69]. This being so, TTR is extensively studied in neurological disease. According to research, overexpression of TTR is discovered in the blood of human patients with adenocarcinoma, squamous carcinoma, and small cell lung cancer. Moreover, TTR is stimulated LLC and B16 melanoma cell proliferation [70]. α-Synuclein consists of 140 amino acids and is involved in the formation of synaptic plasticity. SNCA mutations occur during the protein misfolding and aggregation, leading to the deposition of insoluble protein Lewy bodies [71]. Various studies have reported the discovery of expression of SNCA in melanoma, breast cancer, and ovarian cancer [72,73]. These points could suppose a similarly mechanism of amyloidogenic protein about the tumor environment. The discovery of APP and APLP signaling pathways may lead to novel therapeutic strategies targeting these two bio-entities.

## Figures and Tables

**Figure 1 ijms-22-04999-f001:**
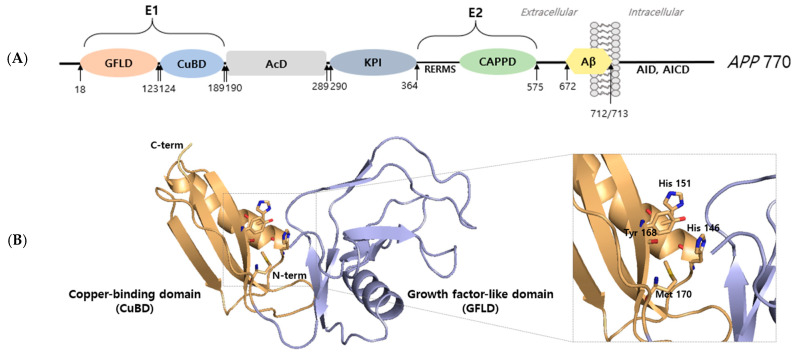
Structural analysis of amyloid precursor protein. (**A**) Schematic representation of the full-length APP 770 isoform domain. Amyloid precursor protein consists of an extracellular 1 (E1) region containing a growth factor-like domain (GFLD) and a copper-binding domain (CuBD). The E1 region is followed by an acidic domain (AcD) and Kunitz protease inhibitor (KPI) domain. The extracellular 2 (E2) region contains the RERMS sequence and the central APP domain (CAPPD). E2 is a linked linker. The region between the linker and transmembrane (TM) constitutes amyloid beta (Aβ). The intracellular region of APP is called APP intracellular domain (AID, AICD). (**B**) X-ray structure (PDB ID: 4PWQ) of the E1 region of APP showing the growth factor-like domain (GFLD) and the copper-binding domain (CuBD). The stick represents the copper binding site.

**Figure 2 ijms-22-04999-f002:**
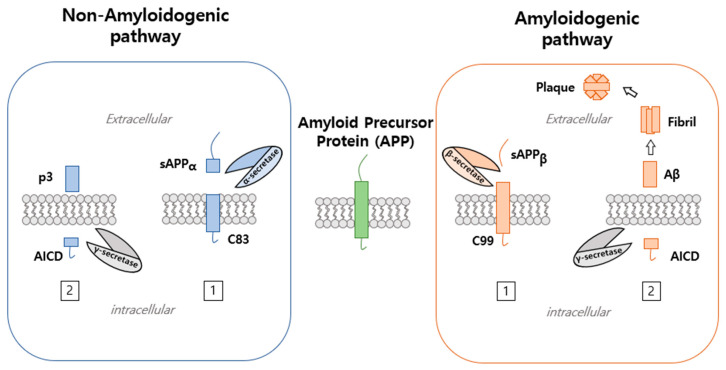
Processing pathway of amyloid precursor protein. APP processing is performed via the non-amyloidogenic or amyloidogenic pathways. The non-amyloidogenic pathway involves APP cleavage by α-secretase and generates soluble amyloid precursor protein cleaved by α-secretase (${\mathrm{sAPP}}_{\alpha }$) and a C-terminal fragment of 83 (C83). This is followed by γ-secretase cleavage, which results in APP non-amyloidogenic extracellular fragment (p3) and APP intracellular domain (AICD). On the other hand, the amyloidogenic pathway involves APP cleaved by β-secretase to generate soluble amyloid precursor protein, which is further cleaved by β-secretase (${\mathrm{sAPP}}_{\beta }$ ) to generate a C-terminal fragment of 99 (C99) that is cleaved by γ-secretase to generate amyloid-beta (Aβ).

**Figure 3 ijms-22-04999-f003:**
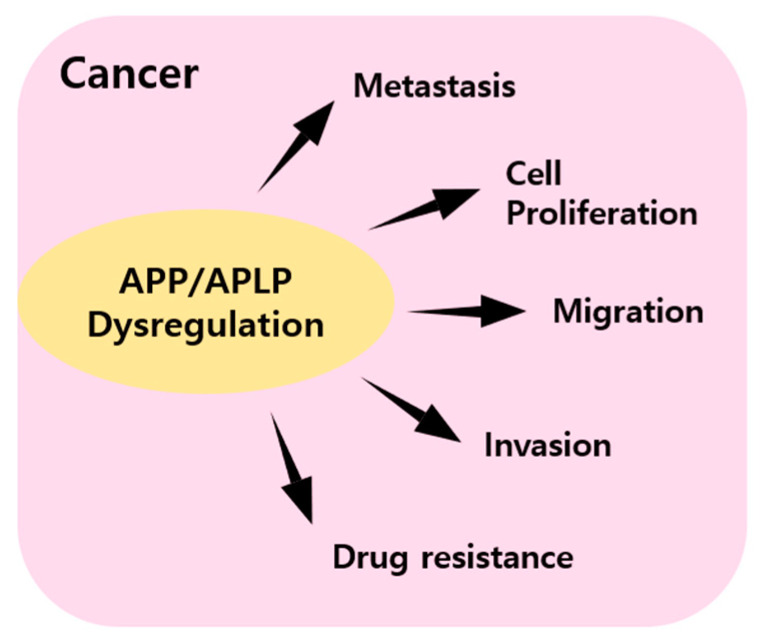
Summary of graphical schemes in which APP and APPLP dysregulation affect cancer metastasis, cell proliferation, migration, invasion, and drug resistance.

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
