# Peer review of "Molecular Characteristics of Amyloid Precursor Protein (APP) and Its Effects in Cancer"

_ijms, 2021, doi:10.3390/ijms22094999_

Round 1
Reviewer 1 Report
I found this manuscript to be a very interesting read. I will write my comments in order of lines.
- Line 12 "Amyloid is written in bold
- Line 33. Replace "cancer" with "cancer cells"
- Line 114. There is a space between "γ" and "secretase"
- Line 122. The authors should write "results in the release of...."
- Line 126. The authors sometimes write sAPPα and other times write APPsα. The wording should be standardized
- Line 192. The authors should decide whether to write "expression" or "overexpression"
- Line 207-209. The concept could be improved
- Line 214. The authors report the association between APP and AR expression in luminal cancer. It would be appropriate to describe the classification of breast cancer (luminal A, luminal B, non luminal) for non-expert readers
- Line 231. The acronym COX-2 should be inserted after cyclooxygenase-2
- Line 259. SUNE-1 cells are human nasopharyngeal carcinoma cell line. Please indicate this information.
- Line 273. "In" is underlined
- Line 307. "Regulation" is written with a capital letter
Author Response
Comments and Suggestions for Authors
I found this manuscript to be a very interesting read. I will write my comments in order of lines.
- Line 12 "Amyloid is written in bold
- -Page 1, line 12: Amyloid precursor protein (APP). It was revised in the text of the paper.
- Line 33. Replace "cancer" with "cancer cells"
- -Page 1, line 33: cancers cells. It was revised in the text of the paper.
- Line 114. There is a space between "γ" and "secretase"
- -Page 3, line 114: γ-secretase complex [4]. It was revised in the text of the paper.
- Line 122. The authors should write "results in the release of...."
- -Page 3, line 122-123: results in the release of both A 40 and A 42 and APP intracellular domain (AID or AICD). It was revised in the text of the paper.
- Line 126. The authors sometimes write sAPPα and other times write APPsα. The wording should be standardized-Page 3, line 125: sAPPα: It was revised in the text of the paper.
- -Page 3, line 108: sAPPβ: It was revised in the text of the paper.
- Line 192. The authors should decide whether to write "expression" or "overexpression"
- -Page5, line191-192: overexpression. It was revised in the text of the paper.
- Line 207-209. The concept could be improved
- -Page5, line 206-209: Breast cancer is divided different subtypes in expression of estrogen receptor (ER), progesterone receptor (PgR), and human epidermal growth factor 2 (HER2). Androgen receptor (AR) is expressed in more than 60% of breast cancer and 90% of ER-positive tumors [34]. It was revised in the text of the paper.
- Line 214. The authors report the association between APP and AR expression in luminal cancer. It would be appropriate to describe the classification of breast cancer (luminal A, luminal B, non luminal) for non-expert readers
- -Page 5, line 214: luminal A. It was revised in the text of the paper.
- Line 231. The acronym COX-2 should be inserted after cyclooxygenase-2
- -Page6, line 230: cyclooxygenase-2 (COX-2), It was revised in the text of the paper.
- Line 259. SUNE-1 cells are human nasopharyngeal carcinoma cell line. Please indicate this information.
- -Page6, line 258: human nasopharyngeal carcinoma cell line (SUNE-1), It was revised in the text of the paper.
- Line 273. "In" is underlined
- -Page7, line 273: in, It was revised in the text of the paper.
- Line 307. "Regulation" is written with a capital letter After modification = The intracellular domain contains a highly conserved sequence that is crucial for the functional regulation of APP, it is believed to mediate interactions with other proteins that lead to the productions of transcriptional factors and initiate nuclear signaling (Figure 1).Before modification = In the Golgi network, both α-secretase and β-secretase can encounter APP that is cleaved α and C99 [21].
- After modification = In the Golgi network, both α-secretase and β-secretase can encounter APP, leading to occur a part of APP proteolysis [21].
- It was revised in the text (Page 4, line 163-164) of the paper.
- Before modification = The intracellular domain contains a highly conserved sequence that is crucial for the functional regulation of APP, whereas its cytoplasmic region is believed to mediate interactions with other proteins that lead to the productions of transcriptional factors and initiate nuclear signaling (Figure 1).
- It was revised in the text (Page 2, line 73-76) of the paper.
- -Page7, line 307: regulation, It was revised in the text of the paper.

Reviewer 2 Report
I have read with interest the current review from Lee and colleagues on the “Molecular characteristics of Amyloid precursor protein (APP) and its effects in cancer”. Overall, the manuscript is well-written and well-structured, and it satisfactorily summarises the current knowledge on the APP association with various types of cancer.
Nevertheless, I have a few comments that I would like the author to address in the revised manuscript.
- Line 192-193: “According to research, epidermal growth factor receptor expression (overexpression?) in breast cancer can influence the central nervous system [30].” The author should clear if whether they mean expression or overexpression.
- I would greatly improve the overall presentation of the review if the authors provided a graphical scheme summarizing the different intracellular/extracellular signalling pathways affected by APP in different types of cancer described in the current manuscript.
- Besides APP, dysregulation of several well-known amyloidogenic proteins such as alpha-synuclein (PMID: 26940507) and transthyretin (PMID: 22471982) have been associated with tumorigenesis/tumour growth (PMID: 30567728; PMID: 26504519; PMID: 33066407). The authors should acknowledge these studies, and highlight potential common mechanisms shared by APP and other aggregation-prone proteins with cancer.
Author Response
Comments and Suggestions for Authors
I have read with interest the current review from Lee and colleagues on the “Molecular characteristics of Amyloid precursor protein (APP) and its effects in cancer”. Overall, the manuscript is well-written and well-structured, and it satisfactorily summarises the current knowledge on the APP association with various types of cancer.
Nevertheless, I have a few comments that I would like the author to address in the revised manuscript.
- Line 192-193: “According to research, epidermal growth factor receptor expression (overexpression?) in breast cancer can influence the central nervous system [30].” The author should clear if whether they mean expression or overexpression.
- -overexpression: It was revised in the text (Page5, line 191-192) of the paper.
- I would greatly improve the overall presentation of the review if the authors provided a graphical scheme summarizing the different intracellular/extracellular signalling pathways affected by APP in different types of cancer described in the current manuscript.Figure 3. Summary of graphical schemes in which APP and APPLP dysregulation affect cancer metastasis, cell proliferation, migration, invasion, and drug resistance.
- Besides APP, dysregulation of several well-known amyloidogenic proteins such as alpha-synuclein (PMID: 26940507) and transthyretin (PMID: 22471982) have been associated with tumorigenesis/tumour growth (PMID: 30567728; PMID: 26504519; PMID: 33066407). The authors should acknowledge these studies, and highlight potential common mechanisms shared by APP and other aggregation-prone proteins with cancer.
- It was revised in the text (Page8, line 350-363) of the paper. Figure 3 was added to the paper and drawn.
- -Besides APP and APLP, another amyloidogenic protein such as α-synuclein (SNCA) and transthyretin (TTR) is reported the association with tumorigenesis and tumor growth. TTR is a tetrameric protein composed of 127 amino acid residue and is synthesized in the liver and choroid plexus of the brain. This protein is occurred single amino acid substitutions, leading on familial amyloidotic polyneuropathy (FAD) [69]. This being so, TTR is extensively studied in the neurological disease. According to research, Overexpression of TTR is discovered in the blood of human patients with adenocarcinoma, squamous carcinoma, and small cell lung cancer. Also, TTR is stimulated LLC and B16 melanoma cell proliferation [70]. SNCA is consist of 140 amino acids that this is involved in the formation of synaptic plasticity. SNCA mutations occurred the protein misfolding and aggregation, leading deposition of insoluble protein-Lewy bodies [71]. Various studies are reported that expression of SNCA is discovered in melanoma, breast cancer, and ovarian cancer [72,73]. These points could suppose similarly mechanism of amyloidogenic protein about tumor environment.

Round 2
Reviewer 2 Report
The current version still has some typos and in errors that should be corrected in the revised manuscript.
Line 355: "(...) leading to familial amyloidotic polyneuropathy (FAD)" Acronym should be: FAP
Line 361: " (...) leading to the deposition of in361 soluble protein-Lewy bodies [71]." Reference [71] does not relate specifically with aSyn aggregation, and thus reference PMID: 26940507 should be added, as suggested in the initial report.
Line 358: "SNCA is consists of 140 amino acids that ...". Typo: it should be "aSynuclein consists"
Author Response
Comments and Suggestions for Authors
The current version still has some typos and in errors that should be corrected in the revised manuscript.
Line 355: "(...) leading to familial amyloidotic polyneuropathy (FAD)" Acronym should be: FAP
Page 8, Line 355: (FAP): It has been revised in the text of the paper.
Line 361: " (...) leading to the deposition of in361 soluble protein-Lewy bodies [71]." Reference [71] does not relate specifically with aSyn aggregation, and thus reference PMID: 26940507 should be added, as suggested in the initial report.
Page 8, Line 361: [71] reference: It has been revised in the text of the paper.
Page 11. Line 521-522: [71] reference
- Lassen, L. B.; Reimer, L.; Ferreira, N.; Betzer, C., & Jensen, P. H. Protein Partners of α‐Synuclein in Health and Disease. Brain pathology 2016, 26, 389-397, doi: 10.1111/bpa.12374: It has been revised in the text of the paper.
Line 358: "SNCA is consists of 140 amino acids that ...". Typo: it should be "aSynuclein consists"
Page 8, Line 358: α-synuclein: It has been revised in the text of the paper.
